# Screening of Candidate Genes Associated with Brown Stripe Resistance in Sugarcane via BSR-seq Analysis

**DOI:** 10.3390/ijms232415500

**Published:** 2022-12-07

**Authors:** Wei Cheng, Zhoutao Wang, Fu Xu, Guilong Lu, Yachun Su, Qibin Wu, Ting Wang, Youxiong Que, Liping Xu

**Affiliations:** National Engineering Research Center for Sugarcane, Key Laboratory of Sugarcane Biology and Genetic Breeding, Ministry of Agriculture and Rural Affairs, College of Agriculture, Fujian Agriculture and Forestry University, Fuzhou 350002, China

**Keywords:** sugarcane, brown stripe resistance, DEGs, BSR-seq, expression profiling, candidate genes

## Abstract

Sugarcane brown stripe (SBS), caused by the fungal pathogen *Helminthosporium stenospilum*, is one of the most serious threats to sugarcane production. However, its outbreaks and epidemics require suitable climatic conditions, resulting in the inefficient improvement of the SBS resistance by phenotype selection. The sugarcane F_1_ population of SBS-resistant YT93-159 × SBS-susceptible ROC22 was used for constructing the bulks. Bulked segregant RNA-seq (BSR-seq) was then performed on the parents YT93-159 (T01) and ROC22 (T02), and the opposite bulks of 30 SBS-susceptible individuals mixed bulk (T03) and 30 SBS-resistant individuals mixed bulk (T04) collected from 287 F_1_ individuals. A total of 170.00 Gb of clean data containing 297,921 SNPs and 70,426 genes were obtained. Differentially expressed genes (DEGs) analysis suggested that 7787 and 5911 DEGs were identified in the parents (T01 vs. T02) and two mixed bulks (T03 vs. T04), respectively. In addition, 25,363 high-quality and credible SNPs were obtained using the genome analysis toolkit GATK for SNP calling. Subsequently, six candidate regions with a total length of 8.72 Mb, which were located in the chromosomes 4B and 7C of sugarcane wild species *Saccharum spontaneum*, were identified, and 279 genes associated with SBS-resistance were annotated by ED algorithm and ΔSNP-index. Furthermore, the expression profiles of candidate genes were verified by quantitative real-time PCR (qRT-PCR) analysis, and the results showed that eight genes (*LRR-RLK*, *DHAR1*, *WRKY7*, *RLK1*, *BLH4*, *AK3*, *CRK34*, and *NDA2*) and seven genes (*WRKY31*, *CIPK2*, *CKA1*, *CDPK6*, *PFK4*, *CBL2*, and *PR2*) of the 20 tested genes were significantly up-regulated in YT93-159 and ROC22, respectively. Finally, a potential molecular mechanism of sugarcane response to *H. stenospilum* infection is illustrate that the activations of ROS signaling, MAPK cascade signaling, Ca^2+^ signaling, ABA signaling, and the ASA-GSH cycle jointly promote the SBS resistance in sugarcane. This study provides abundant gene resources for the SBS resistance breeding in sugarcane.

## 1. Introduction

Sugarcane (*Saccharum* spp. hybrids) is the most important sugar crop worldwide and accounts for 85–90% of sugar production in China. Pests and diseases, reducing the viability and affecting the growth of sugarcane, are the main factors limiting its yield and sugar content [1]. Sugarcane brown stripe (SBS), caused by the infection of *Helminthosporium stenospilum* Drechsler, is a fungal foliar disease [1,2]. Currently, it has become one of the most important diseases in sugarcane planting areas, especially in China, and the severity of the disease is increasing year by year [1]. An outbreak of this disease results in serious yield losses, generally in the range of 18–35% and even up to 40% [1]. Considering the year-long growth cycle, planting resistant varieties is the most environmentally friendly and cost-effective control strategy against diseases in sugarcane [1,3], especially fungal diseases that produce large pathogenic spores, such as SBS.

Modern sugarcane hybrids have a large genome, and their genetic background is highly complex, composed of a different number of chromosomes from *S. officinarum* and *S. spontaneum*, and with a high polyploidy (10×–12×) [4,5,6,7]. The information on the genomic polymorphic sites and different allele frequencies is the key to population genetics research [8]. The rapid development of high-throughput sequencing technology has been accompanied by a significant reduction in sequencing costs. Transcriptome sequencing (RNA-seq) has been widely used in the analysis of differentially expressed genes, the rapid and effective identification of single nucleotide polymorphism (SNP) between different genotypes of the analyzed population, the screening of candidate genes for related traits and the expression analysis of related genes, and the development of molecular markers with high-resolution features that are closely linked to traits [9,10,11].

Bulked segregant RNA-seq (BSR-seq) is a technical method based on the advantages of bulked segregant analysis (BSA) and high-throughput sequencing technology [12,13]. In 1991, the concept of BSA was first put forward by Michelmore et al. [12], which was then successfully used for screening three random amplified polymorphic DNA (RAPD) molecular markers closely linked with the resistance genes *Dm5* and *Dm8* of *Bremia lactucae* in the F_2_ segregating population [12]. In recent years, BSR-seq, with its technical advantages such as high throughput, low cost, and high benefit, has been widely applied in the research of plant disease resistance, especially for crops with complex genetic backgrounds and large genomes. By combining with the reference genome data, this technology has been widely applied, such as for the rapid location of target trait-related genes [14,15], gene mining [16,17], identification of linkage markers [18,19], and fine location on a large scale [17,20,21]. In a word, BSR-seq [22] is an effective and rapid gene mapping technique that combines the advantages of BSA and RNA-seq, and, along with the association analysis of Euclidean distance (ED) [23] and SNP-index [23,24] method, can greatly reduce the workload of marker detection, quickly locate the relevant candidate regions related to the target traits, and then predict the candidate genes.

Sugarcane is an asexually reproductive plant. As far as QTL mapping of target traits is concerned, there is no near population such as doubled haploid (DH) or recombinant inbred line (RIL), and only self-crossing or segregating populations can be used. With the development of DNA molecular marker technology, QTL mapping has developed rapidly as well for different quantitative traits of sugarcane. For example, Lu et al. [25] located 31 QTLs that control chlorophyll content (CC) using a sugarcane F_1_ population generated from a cross between YT9159 and ROC22. You et al. [3] used a self-cross F_1_ population derived from sugarcane variety CP80-1827 and identified 23 QTLs tightly linked to the resistance of sugarcane ratoon stunting disease (*Leifsonia xyli* subsp. *xyli*, *Lxx*). In general, gene mapping is time-consuming and expensive, especially for crops with huge genomes and complex genetic backgrounds. The size of the mapping population, the number, type, and resolution of molecular markers are the main factors that limit gene mapping [18]. Previous studies have shown that the BSA-seq technique is relatively suitable for the initial localization of target traits in species with small genomes and few repetitive sequences in the genomes, such as *Arabidopsis thaliana* [26], *Cucumis sativus* [27], and *Oryza sativa* [28]. However, this rule is assumed to have obvious defects in crops with large genomes, such as wheat (*Triticum aestivum*), maize (*Zea mays*), and sugarcane (*Saccharum* spp. hybrids). At present, BSR-seq has been used to locate major genes and related quantitative traits, and screen candidate genes related to traits in many crops [13].

The present study aimed to screen for differential expression of candidate genes in four different samples associated with brown stripe resistance in sugarcane via BSR-seq. According to the field performance, a total of 30 progenies of extreme resistance/susceptibility to brown stripe were selected from the F_1_ segregating population derived from YT93-159 (SBS-resistant sugarcane variety, 2n = 108) × ROC22 (SBS-susceptible sugarcane variety, 2n = 110), respectively. Secondly, the differential gene expression by RNA-seq and association analysis were performed on all samples. Thirdly, the location of the chromosomal candidate regions associated with the trait for brown stripe resistance, functional annotation, and expression profiling of the candidate genes based on transcriptome data were performed. Finally, the expression patterns of candidate genes were verified by qRT-PCR. This study is expected to provide new genetic resources for the molecular breeding of SBS resistance in sugarcane.

## 2. Results

### 2.1. Field Resistance Survey of Sugarcane Brown Stripe

In this study, a total of 287 individuals in the F_1_ population were investigated. The SBS disease grade of each F_1_ progeny was evaluated across all tested individuals according to the established grading scales. Based on field disease severity surveys in the period 2015–2020, the environmental conditions in 2020 were most suitable to the epidemic of SBS, and the male parent ROC22 was identified as highly susceptible to SBS, while sugarcane female parent YT93-159 showed to be highly resistant. As a result, the individuals used for constructing resistant/susceptible bulks were selected mainly based on the performance of the tested progenies in 2020. Finally, 30 progenies with the extremely SBS-resistant and 30 progenies with the extremely SBS-susceptible were selected to construct resistant- and susceptible-bulk, respectively. Furthermore, the Appendix A shows the frequency distribution for disease severity of SBS among the 287 F_1_ individuals under six habitats. The phenotypic data of the disease severity survey of sugarcane brown stripe in 2020 are shown in Appendix A.

### 2.2. Summary of the BSR-seq and Sequence Alignments

After filtering low-quality reads and adaptors, a total of 170.00 Gb of clean data were generated by RNA-seq, including 17.43 Gb from YT93-159 (T01), 20.14 Gb from ROC22 (T02), 71.21 Gb from the SBS-susceptible bulk (T03), and 61.22 Gb from the SBS-resistant bulk (T04). The percentage of bases with Q30 was more than 93.00% and GC content ranged from 52.27 to 53.00%. Clean data were then individually aligned to the *S. spontaneum* reference genome. The average percentage of sequence alignment ranged from 67.72 to 69.95% (Table 1). These results showed that the RNA-seq data were high quality and could meet the requirements of subsequent BSR-seq analysis.

### 2.3. New Gene Analysis and Differentially Expressed Genes

After BSR-seq and quality control, a total of 70,426 genes were identified, of which 15,083 were differentially expressed genes (DEGs). In addition, 18,532 new genes were discovered by filtering out the sequences whose encoded peptide chains were too short (less than 50 amino acid residues) or contained only a single exon. Among them, the number of DEGs between the parents (T01 vs. T02) was 7787, which included 3747 significantly up-regulated genes and 4040 significantly down-regulated genes (Figure 1A). Then, the number of DEGs between the SBS-resistant bulk and the SBS-susceptible bulk (T03 vs. T04) was 5911, which included 2205 significantly up-regulated genes and 3706 significantly down-regulated genes (Figure 1B). In addition, the identified DEGs were analyzed through hierarchical clustering, and genes with the same or similar expression patterns were clustered. Figure 1C and D showed that the expression patterns of DEGs were different between T01 vs. T02, as compared to those between T03 vs. T04. However, the differences between the two groups were similar, suggesting that DEGs may be closely related to the SBS resistance. Furthermore, a total of 1385 DEGs were identified between T01 vs. T02 and T03 vs. T04 after filtering out the duplicate transcripts, and more differential genes among parents than the mixed bulks, 6402 and 4526 DEGs, were identified, respectively (Figure 1E).

### 2.4. SNP Calling and Analysis of the Candidate Regions for SBS Resistance

A total of 635,094 SNPs, composed of SNPs between parents and those between the two bulks, were finally detected using the genome analysis toolkit GATK [29]. Then, SNPs with multiple allelic loci (0) and a reading support degree < 4 (284,243), which were consistent between the two mixed bulks (107,779), and an inconsistency between the parents or between the two corresponding bulks (217,709), were filtered out. Finally, a total of 25,363 high-quality and highly credible SNPs were obtained for subsequent Euclidean distance (ED) and ΔSNP-index analysis to locate candidate regions associated with SBS resistance in sugarcane.

According to the correlation threshold, a total of 12 SBS-resistance-associated regions were identified by ED algorithm analysis (Figure 2), covering a length of 18.46 Mb, and containing a total of 731 genes, including 24 nonsynonymous mutation genes (Appendix A). Additionally, there were other 10 SBS-resistance associated regions identified by ΔSNP-index analysis (Figure 3). These regions covered a length of 17.84 Mb and contained a total of 752 genes, of which 28 were nonsynonymous mutation genes (Appendix A). The results showed that the number of the associated sites was more by the ED algorithm and the relative distribution density was smaller than that of the ΔSNP-index analysis. Combining the results of association analysis by ED algorithm and ΔSNP-index (Appendix A), six candidate regions associated with the SBS resistance were finally located, which covered a length of 8.72 Mb and contained 384 annotated candidate genes. Among them, there were 17 nonsynonymous mutation SNPs between T01 vs. T02 and 48 between T03 vs. T04. Additionally, a total of 279 candidate coding genes containing 13 nonsynonymous mutant genes were identified by multiple databases, including NR, NT, Swiss-Prot, trEMBL, GO, KEGG, and COG.

### 2.5. Functional Annotation, Enrichment Analysis, and DEG Analysis of the Candidate Genes

To obtain gene function information, this study analyzed the conserved domains of candidate genes using NCBIs conserved domain database (CDD) (https://www.ncbi.nlm.nih.gov/Structure/cdd/wrpsb.cgi, accessed on 15 March 2022) and annotated the homologous function of these genes in *A. thaliana* with the TAIR database [18,30]. Finally, 39 candidate genes associated with SBS resistance were screened. The annotation results, listed in Appendix A, showed that these genes were mainly related to plant growth and development, plant hormone signaling, response to external adversity stress, and plant-pathogen interaction, including protein kinase genes, plant resistance-related genes, transcription factors and key enzyme genes of metabolic pathways. They included protein kinase genes, such as leucine-rich repeat receptor-like kinase (LRR-RLK), calcineurin B-like interacting protein kinase 2/11 (CIPK2/CIPK11), calcineurin B-like protein 2, (CBL2), receptor-like protein kinase 1/7 (RLK1/RLK7), calcium-dependent protein kinase 6 (CDPK6), aspartate kinase 3 (AK3), cysteine-rich receptor-like protein kinase 34 (CRK34); transcription factor genes, such as WRKY7, WRKY13, BLH4, and GATA12; plant-pathogen interaction-related genes, such as pathogenesis-related protein 2 (PR2), flagellin-sensitive 2 (FLS2), and mildew resistance locus O 1 (MLO1); other key enzyme genes of a metabolic pathway, such as glutamine-dependent asparagine synthase 1 (ASN1), NAD(P)H dehydrogenase 2 (NDA2), dehydroascorbate reductase 1 (DHAR1), succinate dehydrogenase 1-2 (SDH1-2), chloroplastic NIFS-like cysteine desulfurase (CPNIFS). In summary, the functional annotation of these candidate genes provides basic information for the subsequent investigation of the related expression patterns, disease resistance mechanisms, and the selection of target genes.

In addition, a total of 279 unigenes annotated in the candidate regions associated with SBS resistance were used for Gene Ontology (GO) and Kyoto Encyclopedia of Genes and Genome (KEGG) enrichment analysis. As shown in Figure 4A, they mainly participated in biological processes, such as immune system processes, metabolic processes, biological regulation, and response to a stimulus; molecular functions, comprising binding, enzyme activity regulation, catalytic activity, transport activity, and transcription factor binding activity; cell components, including cell, organelle, membrane, and macromolecular complex. As shown in Figure 4B, these candidate genes were enriched for a total of 41 KEGG pathways, such as plant-pathogen interaction, RNA transport, starch and sucrose metabolism amino acid biosynthesis, plant hormone signaling, phenylpropane biosynthesis, and terpenoid backbone biosynthesis. The statistical results of the COG classification of genes in the associated regions are shown in Appendix A. The results show that 132 genes in SBS resistance-associated regions were annotated into 21 functional COG categories (e value ≤ 1.0 × 10^−5^). The following top three group categories were found: “general functional predictions only” (25), “transcription” (16), and “signal transduction mechanisms”, and “replication, recombination and repair” (15). In addition, DEGs analysis in the SBS resistance association region (including 279 genes) showed that 49 DEGs were differentially expressed (listed in Appendix A). Among them, the number of DEGs between the parents (T01 vs. T02) was 27, which included 17 significantly up-regulated genes and 10 significantly down-regulated genes. Then, the number of DEGs between the SBS-resistant bulk and the SBS-susceptible bulk (T03 vs. T04) was 22, including five significantly up-regulated genes and 17 significantly down-regulated genes. The results lay the foundation for further research on the molecular mechanism of SBS resistance-related genes and provide valuable information for identifying the crucial genes associated with SBS resistance.

### 2.6. Expression Profiling Analysis of Candidate Genes

Based on the RNA-seq data of four samples from BSR-seq, those candidate genes were differentially expressed in SBS-resistant parent YT93-159 (T01), SBS-susceptible parent ROC22 (T02), SBS-susceptible mixed bulk (T03), and SBS-resistance mixed bulk (T04) (Figure 5A). Among them, the highly expressed genes were *CIPK11*, *DHAR1*, *PR2*, *LRR-RLK*, *CBL2*, *CKA1*, and *MAGL9*, while the genes with low expression were *WRKY7*, *MDN1*, *FLS2*, *SDH1-2*, *MLO1*, *AGC1.5*, and *GLR2* in all four samples. In addition, the expression levels of *PDLP1A*, *PAH1*, *LRR-RLK*, and *PMIR1* in T01 and T03 were higher than those in T02 and T04, while the expression patterns of *CKA1*, *MAGL9*, and *WRKY31* were opposite. Interestingly, the expression levels of both *CIPK11* and *CDPK6* in T02 and T03 were higher than those in T01 and T04, respectively, while the expression patterns of *NDA2*, *KP1*, and *BLH4* were opposite (Figure 5A). Meanwhile, the temporal and spatial expression of the above 39 candidate genes in five different tissues of the sugarcane variety ROC22 is shown in Figure 5B. The results showed that, excluding *PMIR1*, *ACC2*, *GMII*, *FLS2*, *MLO1*, *PDLP1A*, *AK3*, *BLH4*, and *WRKY7*, all the other 30 candidate genes were differentially expressed in all tissues of ROC22. Among them, six genes (*CDPK6*, *CBL2*, *PR2*, *DRH1*, *RLK7*, and *SDH1-2*) were highly expressed, especially *PR2*, *CBL2*, and *CDPK6*, whereas eight genes (*KP1*, *GLR2*, *CIPK2*, *MDN1*, *PDLP1A*, *MAGL9*, *AGC1.5*, and *GCL1*) were lowly expressed. It is worth noting that *WRKY31*, *CRK34*, *CPNIFS*, *CKA1*, and *CIPK11* were highly expressed in leaves, while *CIPK11*, *CDPK6*, *PAH*, and *LRR-RLK* were highly expressed in roots. *DHAR1* was specifically expressed in buds. *DRH1* and *PR2* were highly expressed in stem piths. *MCA2*, *NDA2*, *ASN1*, and *CDPK6* were highly expressed in the stem epidermis, and *MCA2* and *NDA2* were specifically expressed in the stem epidermis (Figure 5B). Overall, 29 and 31 candidate genes were differentially expressed in four samples and different tissues of sugarcane ROC22, respectively.

### 2.7. Validation of Candidate Genes by RT-qPCR

According to the results of the expression profiling of 39 candidate genes in the samples T01 and T02, 20 genes were selected for validation by qRT-PCR. The expression of *KUP9*, *RLK7*, *GATA12*, *CIPK11*, and *FLS2* in T01 and T02 had no significant difference (Figure 6). However, the expression levels of *LRR-RLK*, *DHAR1*, *WRKY7*, *RLK1*, *BLH4*, *AK3*, *CRK34*, and *NDA2* in T01 were significantly higher than those in T02 (*p* < 0.05), but opposite for *WRKY31*, *CIPK2*, *CKA1*, *CDPK6*, *PFK4*, *CBL2*, and *PR2*, which were higher in T02 than in T01 (*p* < 0.05). The above results by qRT-PCR were consistent with their expression profiles in BSR-seq. Besides, the expression patterns of candidate genes were obviously different between SBS-resistant and susceptible sugarcane varieties.

## 3. Discussion

Modern sugarcane cultivars are interspecific hybrids with complex genetic backgrounds, larger genomes, an extremely low recombination rate of excellent genes, and undeciphered genomes. Its breeding relies on large populations with segregating traits and phenotypic selection [3,31]. Therefore, mining crucial genes related to brown stripe resistance can provide target gene resources for the molecular breeding of SBS resistance. However, only limited research has been reported till now, including disease characteristics, control measures, and the isolation and identification of this pathogen [1]. With the rapid development of high-throughput sequencing and biotechnology in recent years, the BSR-seq technique has been widely used to quickly locate and screen genes related to target traits in various crops, such as maize [32], wheat [33], kiwifruit [10], and sugarcane [34,35]. Zhu et al. [36] screened six candidate genes associated with wheat powdery mildew in wheat via BSR-seq. Du et al. [14] performed BSR-seq in maize (*Z. mays*) and identified 18 high-probability SNPs and six candidate genes in response to waterlogging stress. Wang et al. [37] screened 22 genes associated with yellow leaf variation in the tea plant (*Camellia sinensis*) through BSR-seq. In addition, there are many reports that the sugarcane F_1_ population can be used in QTL mapping. For example, You et al. [3] identified 82 candidate genes associated with ratoon stunting disease (*Lxx*) via QTL mapping. Yang et al. [38] carried out QTL mapping in an F_1_ population and screened 1574 potential genes linked to orange rust (*Puccinia kuehnii*) resistance in sugarcane. Besides, Zhang et al. [39] screened 171 candidate genes associated with gummy stem blight (*Didymella bryoniae*) resistance in cucumber via QTL mapping. In the present study, we performed the BSR-seq analysis in four samples, and a total of 39 candidate genes related to SBS resistance were identified (Appendix A). These genes play a crucial role in sugarcane response to SBS through plant-pathogen interaction pathways, Ca^2+^ signaling, ROS signaling, and other resistance-related metabolic pathways. Importantly, our study fully demonstrates that the BSR-seq technique can be successfully used to locate and screen candidate genes for related traits in sugarcane.

Plant resistance is a complex process between the host and the pathogen. Pathogenesis-related proteins (PRs) play an important role in the plant-pathogen interaction [40]. Previous studies have shown that after plants were infected by external pathogens, the expression levels of PRs will be significantly up-regulated, and then the expression of downstream pathogenesis-related protein genes was regulated to defend against pathogen infection, and finally, the plant immune was activated [41]. PR2 belongs to β-1, 3-glucanases, which is a hydrolase that can protect plants when infected with pathogens by destroying the cell walls of fungi [41]. Hou et al. [42] reported that the expression levels of *PR1*, *PR2*, *PR3*, *PR4b*, *PR8*, and *PR-pha* were significantly up-regulated in the resistance reaction mediated by the resistance gene *Xa21* in the late stage of rice (*O. sativa*) inoculated with the pathogenic *Xanthomonas oryzae* pv. *oryzae* (*Xoo*), but the expression of *PR5*, *PR6*, *PR15*, and *PR16* did not change significantly. In this study, one *PR2* gene associated with SBS was rapidly identified, and the expression profiling showed that *PR2* was highly expressed in the root, pith, leaf, epidermis, and the bud of sugarcane, as well as in T01, T02, T03, and T04 (Figure 5). Chu et al. [43] cloned the *ScPR1* gene in sugarcane, and it was differentially expressed in sugarcane root, pith, leaf, epidermis, and buds, and the expression level in resistant varieties was higher than that in susceptible varieties, which was consistent with our results. Furthermore, overexpression of *ScPR1* in *A. thaliana* enhanced the tolerance of the transgenic plants to an infection with the pathogen *Pseudomonas syringae* [43]. The overexpression of *ScPR10* in *Nicotiana benthamiana* by Peng et al. [44] can increase the content of intracellular hydrogen peroxide (H_2_O_2_) and the expression levels of immune genes, thus improving the resistance of *N. benthamiana* to tobacco bacterial wilt (*Ralstonia solanacearum*). The above results provide an important theoretical reference for the functional identification of the sugarcane *PR2* gene detected here in the future.

LRR-RLK in plants is a subfamily of the RLK superfamily, which is involved in regulating the growth and development of plants, external biotic and abiotic stress responses, hormone signaling, and other biological functions [45]. *Xa21* has been clearly reported as an effective bacterial blight disease resistance gene [46], and *ERECTA* has been proven to not only regulate the process of ovule development but also participate in the resistance to bacterial wilt (*R. solanacearum*) in *Arabidopsis*. In addition, BAK1 binds to FLS2 and forms a specific ligand in vivo, thus causing the defense response of plants [47]. At present, there is no report on the functional identification of *RLK* genes in sugarcane. Here in our study, six *RLK* candidate genes (*RLK1*, *RLK7*, *LRR-RLK*, *CKA1*, *AK3*, and *CRK34*) were identified, and qRT-PCR analysis showed that *LRR-RLK*, *CRK34*, *AK3*, and *RLK1* had higher expression in the SBS-resistant parent than in the SBS-susceptible parent (Figure 6). It is thus suspected that these genes may play an important role in the response to the SBS invasion, and the functional elucidation of these genes will be a top priority for future research efforts.

Biotic and abiotic stress can induce plant somatic cells to produce specific Ca^2+^ signals, which are transmitted through channel proteins and transporters on cell membranes, thus forming a specific calcium signaling mechanism in plants [48]. As the second messenger in higher plants, Ca^2+^ plays a vital role in the process of plants adapting to various external stresses [49,50]. In higher plants, Ca^2+^ receptors mainly include CDPKs, CaMs, CBLs, and CMLs. Among them, CaMs, CBLs, and CMLs can only interact with specific proteins, such as the CBL-CIPK signaling network [50,51] to regulate the downstream gene expression in response to adversity stress, including pepper phytophthora blight (*Phytophthora capsici*) [52], wheat stripe rust fungus (*Puccinia striiformis*) [53,54], cold [55], drought [56,57], and heat [55]. A *CIPK* homologue gene, *TaCIPK10,* from wheat, which was rapidly induced by *P. striiformis* inoculation and salicylic acid (SA) treatment, was identified and cloned by Liu et al. [53]. The knockdown of *TaCIPK10* significantly reduced wheat resistance to *P. striiformis*, whereas its overexpression resulted in enhanced resistance of wheat to *P. striiformis*. Poddar et al. [58] identified and characterized a total of 22 *CIPKs* genes from chickpea (*Cicer arietinum*) and revealed that *CaCIPK2* and *CaCIPK11* were up-regulated at one or more time points under drought, ABA, and salt stress. In sugarcane, Su et al. [59] revealed that the expression of *ScCIPK21* was up-regulated under ABA stress; *ScCIPK1*, *−2*, *−15*, *−20*, *−21*, and *−28* were up-regulated under PEG treatment; *ScCIPK1*, *−2*, and *−28* were up-regulated under NaCl stress. Besides, *ScCIPK3* plays a vital role in the induction of immune-related genes after seven days post-inoculation with the tobacco bacteria *R. solanacearum*. In this study, the expression levels of *CIPK2* in T02 were higher than those in T01, but there was no significant difference in the expression levels of *CIPK11* between T01 and T02 (Figure 6). Therefore, these results suggested that the Ca^2+^ receptor proteins CIPK2 and CIPK11 might participate in the Ca^2+^ signaling pathway and enhance resistance to SBS.

Transcription factors can specifically bind with *cis*-acting elements in the upstream promoter region with a special structure to exercise the function of gene regulation and expression at the transcription level, thereby activating or inhibiting the expression and regulating the downstream functional genes [60]. Previous studies have revealed that two maize *WRKY* genes, *ZmWRKY19* and *ZmWRKY53*, were significantly up-regulated by *Aspergillus flavus* inoculation in the resistant maize line TZAR101 [61]. *ZmWRKY53* has also been shown to enhance drought and salt stress [61,62]. Rice *WRKY22*, *WRKY13*, *WRKY47*, *WRKY45-1*, *WRKY45-2*, *WRKY30*, *WRKY53*, *WRKY55*/*WRKY31*, and *WRKY104*/*WRKY89* positively regulated rice resistance to blast (*Magnaporthe oryzae*) [63,64], whereas *WRKY76* negatively regulated rice resistance to *M. oryzae* [65]. In this study, two members of the WRKY family, named *WRKY7* and *WRKY31*, were identified by BSR-seq. Among them, the expression levels of *WRKY31* were high in both the SBS-resistant variety YT93-159 and the SBS-susceptible variety ROC22, while the expression of *WRKY7* in YT93-159 was much higher than that in ROC22 (Figure 6). Wang et al. [66] showed that the expression of the sugarcane *ScWRKY5* gene in smut-resistant varieties YZ01-1413 and LC05-136 was significantly higher than that in smut-susceptible varieties ROC22, YZ03-103, and FN40. Hence, these results suggest that *WRKY7* and *WRKY31* may play a positive regulatory role in sugarcane resistance to brown stripe infection.

A potential molecular mechanism of sugarcane and *H. stenospilum* interaction was depicted (Figure 7). When sugarcane is infected with *H. stenospilum*, pathogen-associated molecular patterns (PAMPs) bind to pattern-recognition receptors (PRRs). It can respond to defensive responses in plants, such as the production of ethylene (ETH), the increase in reactive oxygen species (ROS) expression level in vivo, and then mitogen-activated protein kinases (MAPKs) cascade signal transduction, and the activation of plant hormone signal transduction such as salicylic acid (SA) and abscisic acid (ABA) to regulate the defense response of plants, thereby enhancing the resistance to SBS. When plants are infected by external pathogens, the expression of pathogenesis-related proteins (PRs) in vivo will be significantly up-regulated, and by regulating the expression of downstream disease-related genes, thereby defending against external pathogens, and finally making plants immune. Dehydroascorbate reductase (*DHAR*) is a key enzyme gene that can promote the regeneration of ASA in the ascorbic acid-glutathione (ASA-GSH) cycle, reduce the damage of ROS to plants, and enhance the resistance of plants under the infection by *H. stenospilum*. Upon sugarcane and *H. stenospilum* interaction, Ca^2+^ signal influx is generated within plant somatic cells. Ca^2+^ influx can activate the production of calcium sensor proteins, such as calcineurin B-like proteins (CBLs) and calcium-dependent protein kinases (CDPKs), which in turn activate the NADPH oxidase, and then promote the formation of ROS, and ultimately initiate plant defense responses. In addition, CBL-interacting protein kinases (CIPKs) can only respond to biotic and abiotic stress by interacting with Ca^2+^ sensor CBLs, such as by promoting cell wall reinforcement and regulating downstream resistance gene expression, thereby enhancing the resistance to SBS. In summary, when sugarcane interacts with *H. stenospilum*, the activation of MAPK cascade signaling, SA signaling, ABA signaling, ROS signaling, Ca^2+^ signaling, and the ASA-GSH cycle jointly promote the resistance to sugarcane brown stripe.

## 4. Materials and Methods

### 4.1. Plant Materials

An F_1_ segregating population was developed from a cross of ‘ROC22’ (highly susceptible to SBS) and ‘YT93-159’ (highly resistant to SBS) in 2014, and a total of 287 F_1_ individuals were obtained [18]. Based on field disease severity surveys from 2015 to 2020 under the natural conditions of the field, we found that the environmental conditions in 2020 were most suitable for the SBS epidemic. Hence, the individuals used for constructing SBS-resistant/-susceptible mixed bulks were selected mainly based on the performance of the tested progenies in 2020. In addition, field planting was conducted from a farm at an Experimental Station of the Institute of Sugarcane Science, Longchuan, China (longitude 97°88′ E, latitude 24°25′ N, altitude 951 m), and field design uses the method of randomized complete block design (RCBD) with three replicates. Each plot consisted of a single row with 1.0 m and 1.1 m between rows. ROC22 acted as a protective line in order to increase the stress of SBS, and field management was in accordance with the regular management of local field production.

Considering the lack of grading scales for the identification of disease resistance in sugarcane brown stripe and according to previous studies [31,67], this work has developed a five-grade grading scale, which was used to identify and grade the phenotype of field SBS-resistance (disease severity) in this F_1_ population. The grading scale and diagram were shown in Table 2 and Appendix A, respectively.

For RNA-seq, four groups of samples including two sugarcane parent cultivars, YT93-159 (T01) and ROC22 (T02), and the SBS-susceptible/-resistant mixed bulk (T03/T04) were collected, respectively. According to the results of the disease severity of each progeny in the F_1_ population from the cross of YT93-159 × ROC22, parents and each of the 30 individuals of extreme SBS-resistance and extreme SBS-susceptibility were selected in December 2020 after the natural infection by the pathogen *H. stenospilum*. For the samples of the SBS-resistant parent, and 30 materials selected as the SBS-resistant mixed bulk, the sampling details were as follows: 5 individuals with similar growth vigor and without brown stripe disease symptom were selected, and leaf tissues located in the +2 leaf position and in the mid-leaf blade were collected for samples. Then, approximately 1.0 cm × 0.2 cm leaf tissues were taken via sterilized scissors from the mid-leaf blade, and quickly placed in a 2.0 mL centrifuge tube, and placed on dry ice. It is worth noting that the mass of each sample needs to reach about 10 g. In addition, especially for the SBS-susceptible parent and the SBS-susceptible mixed bulk (30 individuals) materials, the sampling details were as follows: 5 individuals with similar growth vigor and disease severity were selected. Then, half of the diseased spots and the healthy parts adjacent on the leaves were taken (approximately 1.0 cm × 0.2 cm) from the +2 leaf position and the mid-leaf blade when sampling. The mass of each sample needs to reach about 10 g. All samples were frozen in liquid nitrogen immediately after collection in the field, and then these samples were stored at −80 °C freezer pending extraction of total RNA.

### 4.2. RNA Extraction, Library Construction, and RNA Sequencing

Total RNA of ROC22, YT93-159, and 30 SBS-resistant and 30 SBS-susceptible F_1_ individuals was extracted using the Trizol reagent (Invitrogen, Carlsbad, CA, USA) following the manufacturer’s protocol. The concentration and quality of extracted RNA were checked using NanoDrop One (Thermo Fisher Sci, Waltham, MA, USA), and their RNA Integrity Number (RIN) was also analyzed using an Agilent 2100 Bioanalyzer (Agilent Technologies, Waldbronn, Germany). Resistant and susceptible RNA bulks were constructed by separately mixing equal amounts of mRNA from the 30 resistant and susceptible F_1_ individuals, respectively. Two parents, 30 highly SBS-resistant individuals mixed bulk and 30 highly SBS-susceptible individuals mixed bulk, were submitted to Biomarker Technologies Co., Ltd. (Beijing, China) for sequencing.

All samples with a RIN ≥ 7 were regarded as meeting the sequencing standard and cDNA libraries were constructed using a TruSeq RNA sample preparation kit (Illumina RS-122–2101, Illumina, CA, USA) according to the manufacturer’s instructions. The quality of the cDNA libraries was again assessed, and the acceptable cDNA libraries were sequenced on an Illumina HiSeq^TM^ 2500 sequencing platform (Appendix A). The raw data of transcriptome sequencing were deposited in National Genomics Data Center (NGDC), Beijing Institute of Genomics, Chinese Academy of Sciences, under Project PRJCA012967 with Genome Sequence Archive (GSA) number CRA008740 (https://ngdc.cncb.ac.cn/gsa/browse/CRA008740; released on 4 November 2022, accessed on 25 October 2022).

After the sequencing of the cDNA libraries, raw data were filtered, and poor-quality reads were eliminated to obtain high-quality clean data using FastQC and Trimmomatic software [68]. The clean data were then aligned to the *S. spontaneum* AP85-441 reference genome (http://www.life.illinois.edu/ming/downloads/Spontaneum_genome/, accessed on 10 March 2022) using STAR [69] for subsequent SNP calling, BSA association mapping, DEGs analysis, gene functional annotation analysis and GO and KEGG pathway analyses.

### 4.3. New Gene Discovery and Differential Expressed Genes (DEGs) Analysis

Based on the *S. spontaneum* AP85-441 reference genome sequence, the comparable reads were spliced using Cufflinks software and compared with the original genome annotation information to find the original unannotated transcript region and discover new transcripts and genes of the species. Clean reads were aligned to the reference genome, and the expression level was calculated with Fragments per KB of exon per million fragments mapped (FPKM). DESeq R package (version 1.18.0) [70] was then used to conduct differential expression analysis among samples. Fold change ≥ 2 and false discovery rate (FDR) < 0.01 were used as screening criteria. In addition, an effective Benjamini–Hochberg method [71] was used to correct the significant P-value. EBSeq [72] was used for differential analysis, and the differential expressed gene sets between the parents and between the mixed bulks were obtained. Then, a hierarchical cluster analysis was performed to display the differential expression patterns of DEGs.

### 4.4. SNP Calling, BSR-seq Association Analysis

GATK software [29] was used to detect the initial SNP calling. According to the positioning result of clean reads in the reference genome, SNP filtering pretreatment such as mark duplicate removal using Picard software [69], and base recalibration and local realignment using GATK, was conducted. The acquired SNPs were filtered as follows [73]: SNP sites with multiple allelic loci, a read support degree of less than 4, consistent between mixing bulks, and where the susceptible mixed bulk gene was not from susceptible parent, were filtered out. Based on the alleles of SNPs between resistant parent versus susceptible parent, and resistant bulk versus susceptible bulk, SNPs were filtered again to remove missing SNPs in any of the four groups. Finally, high-quality and polymorphic SNPs were obtained.

In this study, the association analysis on the candidate regions of SBS resistance was carried out based on ED [23] and ΔSNP-index [23]. ED is a method for identifying SNP loci with genotype differences between the two mixed bulks using RNA-seq data and calculating the ED value of each site to evaluate the regions associated with the target trait. During the analysis, SNP sites with different genotypes between the two mixed bulks were used to calculate the depth of each base in different mixed bulks. To eliminate background noise, the original ED value was processed to (ED)^2^ and then the ED value was fitted using the SNPNUM method [23] (Appendix A). The median + 3 × standard deviation (SD) of the fitted values for all SNP loci was taken as the correlation threshold (0.07) for association analysis.

The ΔSNP-index is a method for SNPs marker and trait association analysis based on the significant genotype frequency differences between the mixed bulks of target traits. The value of difference is expressed by the ΔSNP-index. In order to eliminate the false-positive loci, the ΔSNP-index was fitted using the SNPNUM method [23]. Finally, the value of ΔSNP-index between parents and the SBS-resistant/susceptible bulk was calculated via the following formula:ΔSNP-index = (SNP-index of resistant parent/resistant bulk) − (SNP-index of susceptible parent/susceptible bulk)

Referring to the method reported by Takagi et al. [74], the threshold for SNP detection was set to 100,000 arranged tests with a confidence level of 99%. SNPs in the candidate regions larger than the ΔSNP-index threshold (0.11) were selected as candidate loci associated with the SBS resistance (Appendix A). In this study, the above two association analysis methods were used to locate the candidate regions associated with brown stripe resistance in sugarcane, and the common nonsynonymous mutant genes obtained by both methods were removed, while the remaining nonsynonymous mutant genes were identified as the candidate genes.

### 4.5. Functional Annotation and Enrichment Analysis

According to the BSR-seq analysis, the candidate regions associated with the resistance of SBS were obtained, and in this region, the candidate genes associated with this trait were screened. Gene functional annotation of the candidate regions was carried out by BLAST [75] against the NR (ftp://ftp.ncbi.nih.gov/blast/db/, accessed on 10 March 2022), NT (https://blast.ncbi.nlm.nih.gov/Blast.cgi, accessed on 10 March 2022), trEMBL (https://www.uniprot.org/blast/, accessed on 10 March 2022), Swiss-Prot (https://www.uniprot.org/, accessed on 10 March 2022), and COG (https://www.ncbi.nlm.nih.gov/research/cog-project/, accessed on 10 March 2022) database with the maximum E-value of 1e-15 [76]. GO and KEGG pathway enrichment analyses were performed to help identify potential function of candidate genes. For GO analysis, GO Term Finder (http://search.cpan.org/dist/GO-TermFinder/, accessed on 10 March 2022) was used to describe the molecular function, biological process, and cellular component of the candidate gene. For KEGG pathway enrichment analysis, the KEGG database (https://www.kegg.jp/kegg/pathway.html, accessed on 10 March 2022) was used to BLAST [75] against the metabolic pathway. Moreover, nonsynonymous SNPs between the two parents (T01 vs. T02) and between the susceptible and resistant bulk (T03 vs. T04) were identified in the candidate regions. Subsequently, the conserved structural domains of the candidate genes from the candidate regions were carried out using the CDD database (https://www.ncbi.nlm.nih.gov/structure/cdd/wrpsb.cgi, accessed on 15 March 2022), and the functions of the candidate genes were predicted according to *Arabidopsis* TAIR database (https://www.arabidopsis.org/Blast/index.jsp, accessed on 15 March 2022).

### 4.6. Expression Profiling of Candidate Genes

To identify the key genes related to the SBS resistance, the expression patterns of the candidate genes were analyzed based on the BSR-seq datasets, and RNA-seq datasets of five different tissues (root, leaf, bud, stem pith, and stem epidermis) of sugarcane variety ROC22 (unpublished), and the expression heat map was drawn based on the FPKM expression values of the candidate genes using TBtools software [77].

### 4.7. qRT-PCR Verification of Candidate Genes

According to the results of expression profiling analysis, 20 candidate genes differentially expressed between the resistant parent (T01) and the susceptible parent (T02) were selected for qRT-PCR verification. According to the instructions of Prime-ScriptTM RT Reagent Kit (TaKaRa, Shimogyo-ku, Kyoto, Japan), first strand cDNA synthesis was performed. The *GAPDH* [78] was chosen as an internal control gene to normalize the expression data. Twenty primer pairs were designed by Primer Premier 6 based on the coding sequences of the 20 selected genes, and their details were shown in Appendix A. PCR amplification was performed in a total volume of 20 μL, containing 10.0 μL FastStart Universal SYBR Green PCR Master (Roche, Shanghai, China), each 0.8 μL (10 µM), forward- and reverse- primers, and 1.0 µL first-strand cDNA template, and makeup to 20 µL with sterile ddH_2_O. The qRT-PCR reactions were performed in ABI QuantStudio^TM^ 3 system (Thermo Fisher Scientific, Waltham, MA, USA) at 50 °C for 2 min pre-amplification; 95 °C for 10 min; followed by 40 cycles of 95 °C for 15 s, 60 °C for 1 min. Three biological replicates for each sample were conducted, and the relative expression level of each candidate gene was calculated from the 2^−∆∆Ct^ value [79]. Histograms were generated by the software GraphPad Prism 6 [30]. The significance analysis of expression levels among different samples was determined by Duncan’s new multiple range test using IBM SPSS Statistics 22.

## 5. Conclusions

In the present study, a field SBS disease severity survey was conducted on 287 F_1_ individuals. We applied the BSR-seq technology to sequence the samples of YT93-159, ROC22, and two opposite mixed bulks. Then, a total of 1385 DEGs were successfully detected between T01 vs. T02 and T03 vs. T04. Six candidate regions spanning 8.72 Mb and associated with the SBS resistance were located on chromosomes 4B and 7C of sugarcane wild species *S. spontaneum*, in which 279 genes including 13 nonsynonymous genes were identified, and 39 candidate genes were selected for in-depth annotation. GO and KEGG enrichment analysis showed that the associated genes in the candidate regions participated in biological processes, such as immune system processes, metabolic processes, biological regulation, and response to external stimuli, and were mostly enriched in plant-pathogen interaction and plant hormone signaling metabolic pathways, suggesting that they could be potentially involved in disease resistance. Furthermore, the expression levels of 20 selected candidate genes were verified via qRT-PCR, and the results were consistent with the data in BSR-seq datasets, indicating the data reliability. Finally, a potential mechanism of molecular interaction between sugarcane and *H. stenospilum* was drawn, suggesting that the activations of ROS signaling, MAPK cascade signaling, Ca^2+^ signaling, ABA signaling, and the ASA-GSH cycle jointly promote the SBS resistance in sugarcane. Our results showed that BSR-seq has the full capacity to screen the candidate genes associated with the target traits, and the present study provides gene resources for sugarcane disease resistance breeding.

## Figures and Tables

**Figure 1 ijms-23-15500-f001:**
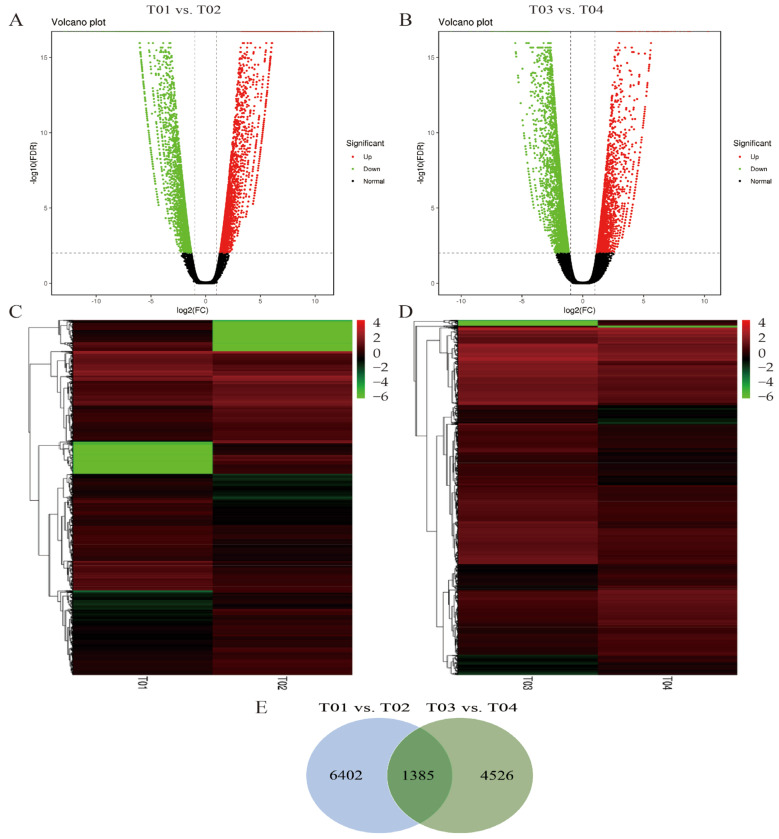
Differentially expressed genes (DEGs) between YT93-159 (T01) and ROC22 (T02), and between the SBS-susceptible bulk (T03) and SBS-resistant bulk (T04). Volcano plot analysis of DEGs for the pairwise comparisons T01 vs. T02 (**A**) and T03 vs. T04 (**B**). Hierarchical cluster analysis of DEGs for the pairwise comparisons T01 vs. T02 (**C**) and T03 vs. T04 (**D**). Wayne map analysis of DEGs between the pairwise comparisons T01 vs. T02 and T03 vs. T04 (**E**).

**Figure 2 ijms-23-15500-f002:**
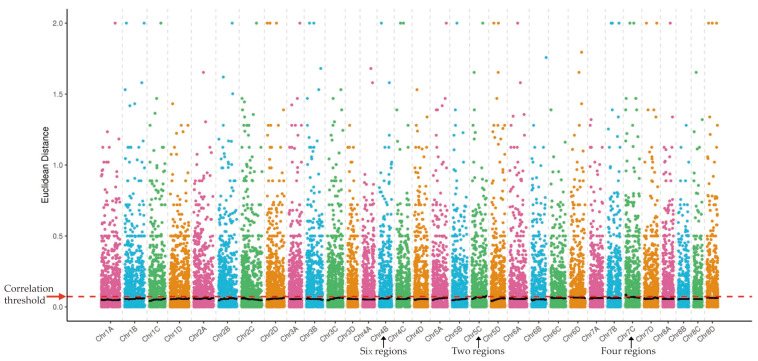
The distribution of correlation thresholds of Euclidean distance (ED) algorithm on the chromosome of *Saccharum spontaneum*. The abscissa is the chromosome name and the ordinate is the ED value; the colored dots represent the ED value of each SNP locus on the chromosome, the black line represents the fitted ED value, and the red arrow represents the correlation threshold.

**Figure 3 ijms-23-15500-f003:**
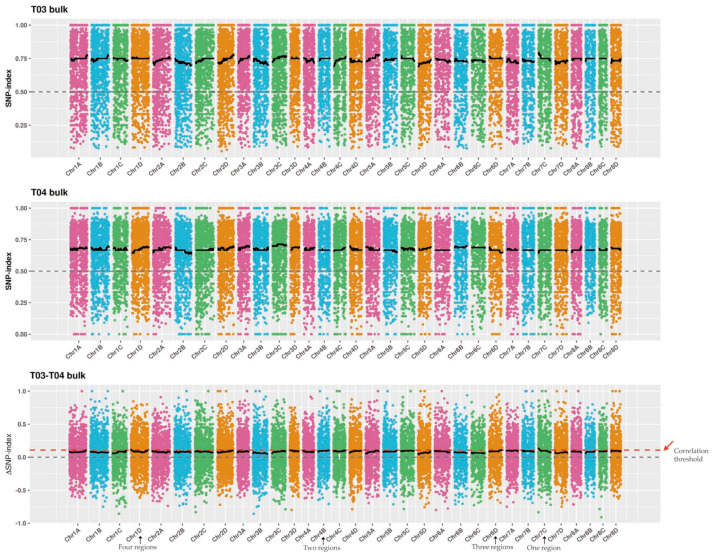
The distribution of correlation thresholds of ΔSNP-index on the chromosome of *Saccharum spontaneum*. The abscissa is the chromosome name, and the ordinate is the SNP-index value (or ΔSNP-index value); the colored dots represent the SNP-index value (or ΔSNP-index value) on the chromosome, the black line represents the fitted SNP-index value (or ΔSNP-index value), and the red arrow represents the correlation threshold.

**Figure 4 ijms-23-15500-f004:**
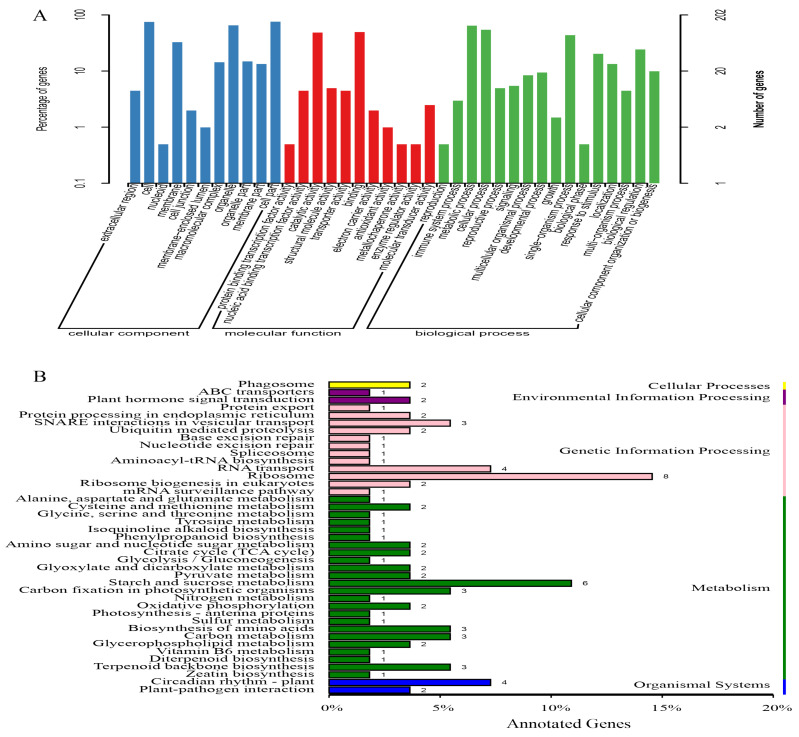
Enrichment analysis of candidate genes in sugarcane brown stripe (SBS) resistance-associated regions. (**A**) GO functional enrichment analysis. (**B**) KEGG pathway enrichment analysis.

**Figure 5 ijms-23-15500-f005:**
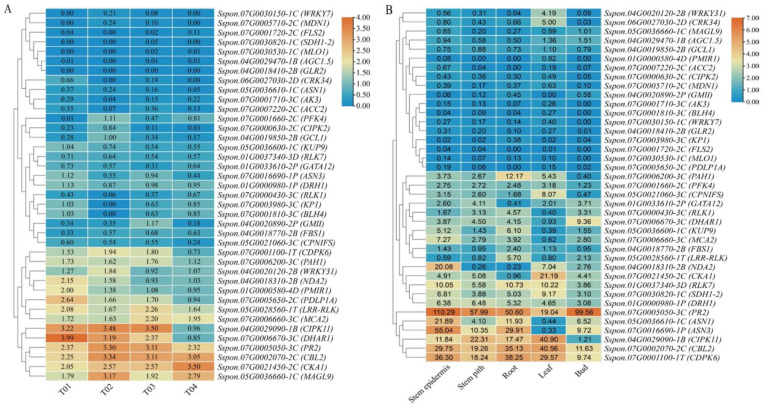
Expression profiles of the candidate genes. (**A**) Expression patterns of 39 candidate genes in four sequenced samples. (**B**) Expression patterns of 39 candidate genes in different tissues of sugarcane parent ROC22. Heatmaps are generated based on FPKM values using TBtools. The color scale represents the range of FPKM values, in which orange and blue represent high- and low-level expression, respectively.

**Figure 6 ijms-23-15500-f006:**
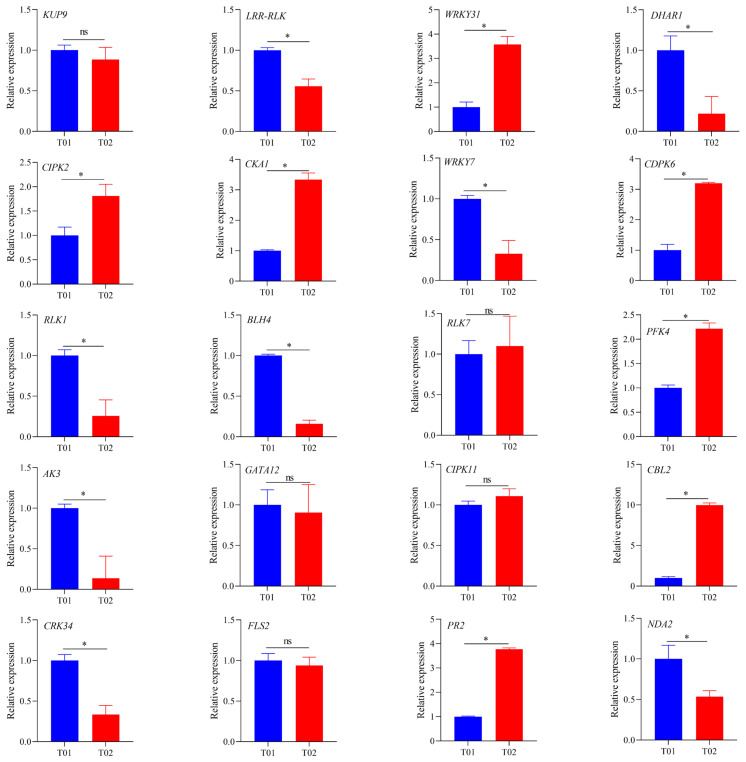
The relative expression of 20 candidate genes by qRT-PCR. The experimental expression level data are normalized to the expression level of the internal reference gene glyceraldehyde-3-phosphate dehydrogenase (*GAPDH*). T01 (blue) and T02 (red) represent the SBS-resistant parent YT93-159 and the SBS-susceptible parent ROC22, respectively. The bar is the standard error of each group of processing data (n = 3). * Duncan’s new multiple range test indicates that candidate genes are significantly differentially expressed at the level of *p* < 0.05 between T01 and T02. ns = Not significant.

**Figure 7 ijms-23-15500-f007:**
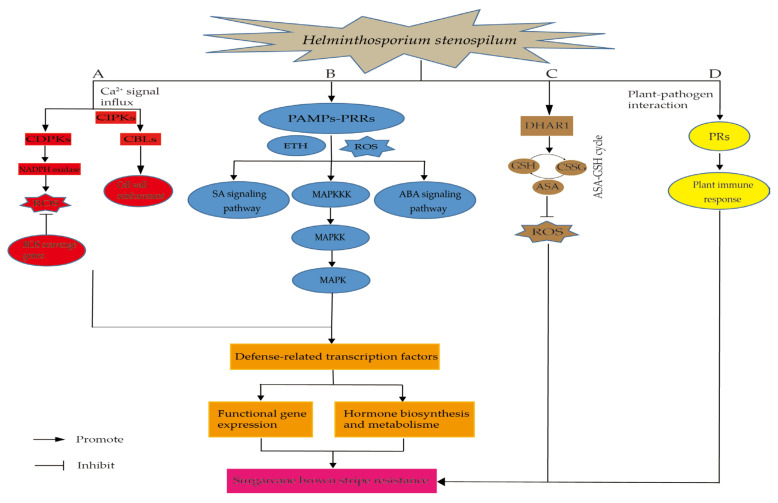
A potential molecular mechanism of sugarcane and *H. stensopilum* interaction. A, Ca^2+^ signaling pathway; B, MAPKs cascade signaling pathway; C, ASA-GSH cycle; D, plant immune system.

**Table 1 ijms-23-15500-t001:** Summary of BSR-seq of four samples.

Items	SBS-Resistant Parent YT93-159 (T01)	SBS-Susceptible Parent ROC22 (T02)	SBS-Susceptible Bulk (T03)	SBS-Resistant Bulk (T04)
Clean bases ^a^	17,433,975,290	20,136,050,680	71,212,242,046	61,215,473,752
Clean reads ^b^	58,329,199	67,443,343	239,023,714	205,084,682
Q30 (%) ^c^	93.33	93.06	94.05	94.25
GC (%) ^d^	52.27	53.00	52.84	52.55
Total reads ^e^	116,658,398	134,886,686	478,047,428	410,169,364
Mapped reads ^f^	81,605,146 (69.95%) ^g^	92,357,559 (68.47%)	323,730,767 (67.72%)	286,908,836 (69.95%)
Unique mapped reads ^h^	39,889,476 (34.19%) ^i^	44,895,323 (33.28%)	166,031,678 (34.73%)	150,882,351 (36.79%)

^a–i^ represents the number of filtered bases; the number of filtered reads; the percentage of bases with a quality value greater than or equal to 30 to the total number of bases; the GC content of the sample; the number of clean reads; The number of reads aligned to the reference genome; the percentage in clean reads; the number of reads aligned to the unique position of the reference genome; the percentage in clean reads, respectively.

**Table 2 ijms-23-15500-t002:** Grading scale for field survey of sugarcane brown stripe (SBS) infected by *H. stenospilum* in sugarcane.

Tarit	Grade	Symptom Description
SBS-resistance	1	No disease symptom on the surface of sugarcane leaves
	2	Disease symptoms appear sporadically on the surface of sugarcane leaves
	3	Mild occurrence. The details are as follows: no more than 1/3 of the sugarcane leaves with disease symptoms, and the upper young leaves (+1, +2, +3 leaves) have no disease symptoms or sporadic disease symptoms
	4	Moderate occurrence. The details are as follows: more than 1/3 of the sugarcane leaves with disease symptoms, and some disease symptoms appear in the upper young leaves (+1, +2, +3 leaves)
	5	Severe occurrence. The details are as follows: more than 1/3 of the sugarcane leaves with disease symptoms, and the upper young leaves (+1, +2, +3 leaves) have more disease symptoms or more than 2/3 of the sugarcane leaves with disease symptoms, and the upper young leaves (+1, +2, +3 leaves) have some disease symptoms

## Data Availability

Not applicable.

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
