# Peer review of "Screening of Candidate Genes Associated with Brown Stripe Resistance in Sugarcane via BSR-seq Analysis"

_ijms, 2022, doi:10.3390/ijms232415500_

Round 1
Reviewer 1 Report
In this study, the sugarcane F1 population of SBS-resistant YT93-159 × SBS-susceptible ROC22 was used for constructing the bulks. Bulked segregant RNA-seq (BSR-Seq) was performed on the parents YT93-159 (T01) and ROC22 (T02), and the opposite bulks of 30 SBS-susceptible bulks (T03) and 30 SBS-resistant bulks (T04) collected from 287 F1 progenies. The differentially expressed genes (DEGs) and SNPs of two parent groups (T01 vs. T02) and two mixed bulks (T03 vs. T04) were analyzed. Six candidate regions spanning 8.72 Mb and associated with the SBS-resistance were located on the chromosomes 4B and 7C of sugarcane wild species S. spontaneum, in which 279 genes including 13 nonsynonymous genes were identified, and 39 candidate genes were screened for in-depth annotation. The expression profiles of candidate genes were verified by quantitative real-time PCR (qRT-PCR) analysis, and the results showed that eight genes (LRR-RLK, DHAR1, WRKY7, RLK1, BLH4, AK3, CRK34, and NDA2) and seven genes (WRKY31, CIPK2, CKA1, CDPK6, PFK4, CBL2, and PR2) of the 20 tested genes were significantly up-regulated in YT93-159 and ROC22, respectively. The results of this study were innovative and provides abundant gene resources for the SBS resistance breeding in sugarcane, it has important significance and application value for the prevention and control of SBS in sugarcane. The manuscript could publication in ‘International Journal of Molecular Sciences’ after revised.
1. Line 40, the full name of the pathogen should be "Helminthosporium stenospilum Drechsler" when it first appears in the manuscript, and the abbreviation should be changed to "H. stenospilum".
2. Lines 70-83, the introduction of BSR-Seq technology is too long, and it is recommended to simplify.
3. Lines 94-103, insert “[13, 22, 28, 29]” behind “in many crops”, and remove this sentence: “Based on BSR-Seq and the F2 segregation population, Liu et al. [13] located the maize waxy gene gl3 in a region of about 2 Mb on chromosome 4, and a transcription factor gene MYB was discovered as a candidate gene for this trait. Lin et al [28] screened 24 candidate genes for low-temperature tolerance in Actinidia arguta by BSR-Seq. With the same method, Hou et al. [22] identified seven candidate genes related to flowering time traits in Paeonia suffruticosa. In addition, Gao et al. [29] located four correlation intervals of target traits via BSR-Seq and found four candidate genes for sugarcane smut resistance through the annotation of functional genes in candidate regions. As far as SBS is concerned, there has been no research report involving the mining of candidate resistance genes for SBS resistance”.
4. Lines 106-109, remove this sentence: “Firstly, the field disease severity identification and evaluation were conducted from 2015-2020, especially the phenotypic performance of brown stripe disease in 2020, due to the reason that the outbreaks and the highest severity of SBS was appeared in 2020, and the paternal parent ROC22 showed to be highly susceptible in 2020. Hence,”. And line 109, “according” should be changed to “According”.
5. Line 184, “24 were nonsynonymous mutation genes” or “28 were nonsynonymous mutation genes”? Please check it!
6. Line 268, “29 and 31 candidate genes” or “39 and 30 candidate genes”? Please check it!
7. Lines 282-283, “but opposite for WRKY31, CKA1, CDPK6, PFK4, CBL2 and PR2, higher in T02 than in T01 (P < 0.05).”, there are 6 genes. However, line29, in the abstract: “and seven genes (WRKY31, CIPK2, CKA1, CDPK6, PFK4, CBL2, and PR2) of the 20 tested genes were significantly up-regulated in YT93-159 and ROC22, respectively” and in Figure 5, there are 7 genes significantly up-regulated? Please check it!
8. Line 437, samples are placed on dry ice, while in line 443-444, all samples are frozen in liquid nitrogen immediately after collection?
9. Lines 438 and 443, "10 grams" should be changed to "10 g".
10. Lines 454-456, “A total of 62 RNA samples, including two parents, 30 highly SBS-resistant progenies and 30 highly SBS-susceptible progenies, were prepared and submitted to Biomarker Technologies Co., Ltd. (Beijing, China) for sequencing” should be changed to “Two parents, 30 highly SBS-resistant mixed bulks and 30 highly SBS-susceptible mixed bulks, were submitted to Biomarker Technologies Co., Ltd. (Beijing, China) for sequencing”.

Author Response
Point to point response to reviewer 1
Dear reviewer,
We are glad to receive your valuable comments and suggestions to our manuscript. Thank you for your kind consideration on this manuscript "Screening of candidate genes associated with brown stripe resistance in sugarcane via BSR-Seq analysis". Without your professional reviews, this manuscript would not be as smooth and more persuasive as what it is now.
We have amended the manuscript according to all the opinions, suggestions and comments of the reviewers and all the changes have been marked-up in the text by the red or blue fond. The responses to all the comments and suggestions are itemized as follows:
Comment 1: Line 40, the full name of the pathogen should be "Helminthosporium stenospilum Drechsler" when it first appears in the manuscript, and the abbreviation should be changed to "H. stenospilum".
Revision: Thanks for your valuable comment. We confirmed the scientific name of the SBS pathogen, and Line 14, Line 41, and Line 414 shows the revised content.
Comment 2: Lines 70-83, the introduction of BSR-Seq technology is too long, and it is recommended to simplify.
Revision: Thanks for your comment. According to your professional suggestion, Lines 72-77 show the simplified content about the introduction of BSR-Seq technology.
Comment 3: Lines 94-103, insert “[13, 22, 28, 29]” behind “in many crops”, and remove this sentence: “Based on BSR-Seq and the F2 segregation population, Liu et al. [13] located the maize waxy gene gl3 in a region of about 2 Mb on chromosome 4, and a transcription factor gene MYB was discovered as a candidate gene for this trait. Lin et al [28] screened 24 candidate genes for low-temperature tolerance in Actinidia arguta by BSR-Seq. With the same method, Hou et al. [22] identified seven candidate genes related to flowering time traits in Paeonia suffruticosa. In addition, Gao et al. [29] located four correlation intervals of target traits via BSR-Seq and found four candidate genes for sugarcane smut resistance through the annotation of functional genes in candidate regions. As far as SBS is concerned, there has been no research report involving the mining of candidate resistance genes for SBS resistance”.
Revision: Thanks for your professional suggestion. We revised it accordingly at the Lines 88.
Comment 4: Lines 106-109, remove this sentence: “Firstly, the field disease severity identification and evaluation were conducted from 2015-2020, especially the phenotypic performance of brown stripe disease in 2020, due to the reason that the outbreaks and the highest severity of SBS was appeared in 2020, and the paternal parent ROC22 showed to be highly susceptible in 2020. Hence,”. And line 109, “according” should be changed to “According”.
Revision: Thanks for this comment. According to your valuable suggestions, we revised this sentence (Line 91).
Comment 5: Line 184, “24 were nonsynonymous mutation genes” or “28 were nonsynonymous mutation genes”? Please check it!
Revision: Thanks for this comment. This is the author's carelessness. The correct description is as follows: Line 166, “24 were nonsynonymous mutation genes”, and Line 169, “28 were nonsynonymous mutation genes”. We have revised it.
Comment 6: Line 268, “29 and 31 candidate genes” or “39 and 30 candidate genes”? Please check it!
Revision: Thanks for your comment. After many times inspections, there are not errors. The expression profiling results further showed that among all 39 genes, 29 and 31 candidate genes were differentially expressed in four sequenced samples and different tissues of sugarcane ROC22, respectively.
Comment 7: Lines 282-283, “but opposite for WRKY31, CKA1, CDPK6, PFK4, CBL2 and PR2, higher in T02 than in T01 (P < 0.05).”, there are 6 genes. However, line29, in the abstract: “and seven genes (WRKY31, CIPK2, CKA1, CDPK6, PFK4, CBL2, and PR2) of the 20 tested genes were significantly up-regulated in YT93-159 and ROC22, respectively” and in Figure 5, there are 7 genes significantly up-regulated? Please check it!
Revision: Thanks for your valuable comment. We left out the CIPK2 gene, and now it has been added at Line 268.
Comment 8: Line 437, samples are placed on dry ice, while in line 443-444, all samples are frozen in liquid nitrogen immediately after collection?.
Revision: Thanks for this comment. In order to minimize the degradation of samples when we collect RNA-seq samples, we first sample on dry ice, and frozen in liquid nitrogen immediately after sampling.
Comment 9: Lines 438 and 443, "10 grams" should be changed to "10 g".
Revision: Thanks for your comment. According to your comment, it has been modified accordingly at Line 421 and Line427.
Comment 10: Lines 454-456, “A total of 62 RNA samples, including two parents, 30 highly SBS-resistant progenies and 30 highly SBS-susceptible progenies, were prepared and submitted to Biomarker Technologies Co., Ltd. (Beijing, China) for sequencing” should be changed to “Two parents, 30 highly SBS-resistant mixed bulks and 30 highly SBS-susceptible mixed bulks, were submitted to Biomarker Technologies Co., Ltd. (Beijing, China) for sequencing”..
Revision: Thanks for your comment. According to your professional suggestion, we have revised the sentence at Lines 438-440.
Take this opportunity, again we do want to express our high and great appreciation for your kind words and professional suggestions, which should no doubt help for our future research. Thanks again.
Any questions, we will be more than happy to answer. Looking forward to hearing from you soon.
Regards and Best wishes!
Youxiong Que and Liping Xu
2022-11-24
Reviewer 2 Report
The authors reported the candidate genes associated with brown stripe resistance in sugarcane via BSR-Seq analysis. The results provides the gene resources for the SBS resistance breeding in sugarcane. Some minor revision need to be followed.
(1) Modern sugarcane hybrids have a large genome, and with a highly polyploidy (10×- 12×). The authors should introduce the genetic information of YT93-159 and ROC22.
(2) In 4.1. Plant Materials, Can the authors provide the brown stripe disease infection experiment design information, how many individuals in one biological repeat? how many biological repeats?
(3) The authors should calculate the frequency distribution of the grading scales of disease resistance in F1 segregating population.
(4) Can the authors provide some examples about QTL mapping of F1 segregating population in Introduction or Discussion section?
Author Response
Point to point response to reviewer 2
Dear reviewer,
We are glad to receive your valuable comments and suggestions to our manuscript. Thank you for your kind consideration on this manuscript "Screening of candidate genes associated with brown stripe resistance in sugarcane via BSR-Seq analysis". Without your professional reviews, this manuscript would not be as smooth and more persuasive as what it is now.
We have amended the manuscript according to all the opinions, suggestions and comments of the reviewers and all the changes have been marked-up in the text by the red or blue fond or blue fond. The responses to all the comments and suggestions are itemized as follows:
Comment 1: Modern sugarcane hybrids have a large genome, and with a highly polyploidy (10×- 12×). The authors should introduce the genetic information of YT93-159 and ROC22.
Revision: Thanks for your valuable comment. We have added genetic background information for modern cultivars YT93-159 and ROC22 at Lines 93-94.
Comment 2: In 4.1. Plant Materials, Can the authors provide the brown stripe disease infection experiment design information, how many individuals in one biological repeat? how many biological repeats?
Revision: Thanks for your comment. The F1 segregating population contains two parents and 285 individuals, for a total of 287 lines (Line 389). Hence, 287 individuals in one biological repeat, and field design uses the method of randomized complete block design (RCBD) with three replicates (Lines 396-397).
Comment 3: The authors should calculate the frequency distribution of the grading scales of disease resistance in F1 segregating population.
Revision: Thanks for your professional suggestion. In 2.1, we revised it at Lines 112-114, and the frequency distribution of the grading scales of disease resistance in F1 segregating population were shown in Supplementary Figure S1.
Comment 4: Can the authors provide some examples about QTL mapping of F1 segregating population in Introduction or Discussion section?
Revision: Thanks for your valuable suggestions. In fact, this study is based on BSR-Seq analysis to screen SBS resistance candidate genes, and the previous research work cited in the introduction and discussion of the manuscript is based on BSR-Seq analysis. Due to the difference of analysis methods, if the previous research results of QTL mapping were cited, the author thinks it will appear abrupt and should bring confusion to readers. This is the author's understanding. If the reviewer insists on your comments, the author will make corresponding amendments.
Take this opportunity, again we do want to express our high and great appreciation for your kind words and professional suggestions, which should no doubt help for our future research. Thanks again.
Any questions, we will be more than happy to answer. Looking forward to hearing from you soon.
Regards and Best wishes!
Youxiong Que and Liping Xu
2022-11-24
Reviewer 3 Report
Sugarcane brown stripe is one of the most serious threats to sugarcane production. Using SBS resistance genes to develop resistant varieties is the most economic way to decrease the loss caused by this disease. In this study, BSR-Seq was performed among the parents, resistant and susceptible bulks with differentially expressed genes and 25,363 SNPs were identified between the two parents and two bulks using a F1 segregation population developed by crossing YT93-159 and ROC22. After review of this manuscript, I have some comments as below:
1. The most valuable point of this project is to identify genomic regions harboring SBS resistance genes using BSR-seq, however, in this study, results of this part is not well displayed. The authors claimed that “According to the correlation threshold, a total of 12 SBS-resistance associated regions were identified by ED algorithm analysis (Fig.2A)” and “other 10 SBS-resistance associated regions identified by ΔSNP-index analysis (Figure 2B)”. I checked Figure 2, it is hard to see where those 12 SBS-resistance associated regions identified by ED algorithm analysis (Fig.2A) and the other 10 SBS-resistance associated regions identified by ΔSNP-index analysis (Figure 2B). Besides, the quality of this figure is too low.
2. After the identification of the DEGs and the genomic regions for SBS resistance. It should be focused to validate the regions harboring SBS resistance, not to validate the expression level of those so-called candidate genes. I strongly suggest to add the validation of those regions.
3. It is claimed in method “and field design uses the method of randomized complete block design (RCBD) with three replicates”. I am confusing that the F1 segregation population consists only individual plants, how can you arrange randomized complete block design and replicates?
4. The schematic diagram of BSR-seq is what everyone knows, is it worth putting figure 6 in the manuscript?
5. There are many grammar, format, spelling and expression mistakes across the manuscript. The quality of this manuscript is too low, the English is required to be well edited.
Author Response
Point to point response to Reviewer 3
Dear reviewer,
We are glad to receive your valuable comments and suggestions to our manuscript. Thank you for your kind consideration on this manuscript "Screening of candidate genes associated with brown stripe resistance in sugarcane via BSR-Seq analysis". Without your professional reviews, this manuscript would not be as smooth and more persuasive as what it is now.
We have amended the manuscript according to all the opinions, suggestions and comments of the reviewers and all the changes have been marked-up in the text by the red or blue fond. The responses to all the comments and suggestions are itemized as follows:
Comment 1: The most valuable point of this project is to identify genomic regions harboring SBS resistance genes using BSR-seq, however, in this study, results of this part is not well displayed. The authors claimed that “According to the correlation threshold, a total of 12 SBS-resistance associated regions were identified by ED algorithm analysis (Fig.2A)” and “other 10 SBS-resistance associated regions identified by ΔSNP-index analysis (Figure 2B)”. I checked Figure 2, it is hard to see where those 12 SBS-resistance associated regions identified by ED algorithm analysis (Fig.2A) and the other 10 SBS-resistance associated regions identified by ΔSNP-index analysis (Figure 2B). Besides, the quality of this figure is too low.
Revision: Thanks for your valuable comment. Accordingly, in sections 2.4 and 4.4, we provide a detailed, systematic introduction and analysis of SNP calling, ED algorithm, and ΔSNP-index. For the convenience of readers' understanding of the methods of ED and ΔSNP-index, we have made minor modifications in 4.4, Lines 488-489. Besides, in Line 491 and Line 505, we have added Supplementary Table S6, which helps to show the relevant values in detail. To eliminate background noise in the process of bioinformatics analysis, the original ED value was processed to (ED)2 and then the ED value was fitted using the SNPNUM method. The median + 3 × standard deviation (SD) of the fitted values for all SNP loci was taken as the correlation threshold for association analysis, and 0.07 was calculated. Hence, Figure 2A was generated. In addition, the value of ΔSNP-index is obtained by the following formula: ΔSNP-index = (SNP-index of resistant parent/ resistant bulk) - (SNP-index of susceptible parent/susceptible bulk). In order to eliminate the false-positive loci, the ΔSNP-index was fitted using the SNPNUM method. Referring to the method reported by Takagi et al. [73], the threshold for SNP detection was set to 100,000 arranged tests with a confidence level of 99% (0.11). Hence, Figure 2B was generated, and we have revised a Figure 2 with DPI = 500. Thanks again for your professional advice.
Comment 2: After the identification of the DEGs and the genomic regions for SBS resistance. It should be focused to validate the regions harboring SBS resistance, not to validate the expression level of those so-called candidate genes. I strongly suggest to add the validation of those regions.
Revision: Thanks for your valuable comment. From RNA-seq to obtaining DEGs, from obtaining candidate regions associated with SBS resistance to gene annotation, from in-depth annotation, expression profiling analysis to qRT-PCR validation, the above series of analysis processes are aimed at obtaining the most reliable candidate genes for SBS resistance. After RNA-Seq and quality control, a total of 15,083 were DEGs. Among them, the number of DEGs between the two parents (T01 vs. T02) was 7,787, which included 3,747 significantly up-regulated. In addition, the number of DEGs between the SBS-resistant bulks and the SBS-susceptible bulks (T03 vs. T04) was 5,911, which included 2,205 significantly up-regulated. Obviously, the number of these genes is huge. Hence, we use BSR-Seq technique later. We obtained candidate regions associated with SBS resistance via BSR-Seq analysis, then functionally annotated the candidate regions, and obtained 279 candidate genes. However, this is still a large number of genes. Therefore, we use the CDD database and the TAIR database for in-depth annotation. In the end, 39 candidate genes were obtained. Then, based on transcriptome data, we performed the analysis of expression profiling on these 39 candidate genes, and based on these results, we selected 20 candidate genes to verify their expression patterns in SBS resistant/susceptible parents by qRT-PCR. In summary, we analyzed, filtered, and annotated all genes within the obtained candidate region associated with SBS resistance, according to your suggestion that the present study should be focused to validate the regions harboring SBS resistance. Thanks again and we hope that now it is presented more clearly in the revised manuscript.
Comment 3: It is claimed in method “and field design uses the method of randomized complete block design (RCBD) with three replicates”. I am confusing that the F1 segregation population consists only individual plants, how can you arrange randomized complete block design and replicates?
Revision: Thanks for your professional comment. In response to your doubts, like potato (Solanum tuberosum), sugarcane is an asexually reproducing crop. Regarding how and why it is reasonable to arrange randomized complete block design and replicates in the F1 segregation population consisting only individual plants, the rationality hidden behind was clearly explained in a research paper published in ‘the Crop Journal’ in August 2022, which uses the same F1 population and the same field design. Link to this research paper: https://doi.org/10.1016/j.cj.2021.11.009, Volume 10, Issue 4, Pages 1131-1140.
Comment 4: The schematic diagram of BSR-seq is what everyone knows, is it worth putting figure 6 in the manuscript?
Revision: Thanks for this comment. The reason why this schematic diagram is included in the manuscript is that it is convenient for readers to see at a glance the population design, parent selection, population size, sample number and other information of this study. If needed, I can move it into the supplementary files.
Comment 5: There are many grammar, format, spelling and expression mistakes across the manuscript. The quality of this manuscript is too low, the English is required to be well edited.
Revision: Thanks for your valuable comment. Here we want to confirm that during the revision, we have asked for help from several foreign friends, whose also focused on crop breeding and their mother language is all English, to improve the language expression throughout this manuscript. We are confident that now the language should be smooth.
Take this opportunity, again we do want to express our high and great appreciation for your kind words and professional suggestions, which should no doubt help for our future research. Thanks again.
Any questions, we will be more than happy to answer. Looking forward to hearing from you soon.
Regards and Best wishes!
Youxiong Que and Liping Xu
2022-11-24
Reviewer 4 Report
Studies conducted by Cheng et al., investigates the genomic regions contributing to disease resistance in sugarcane. Overall, the study provide enough of evidence to prove their study. I have following concerns ..
1. Though authors investigated the gene expression profile of the resistance and susceptible line and tracked down the possible candidate genes, I did not see any analysis showing the difference in genomic content of these lines. or the genomic variation. I would suggest the authors to check all the possible genes individuaily and compare the polymorphism in these genes and compare between resistance and susceptible varieties.
2. Authors should check the haplotypic diversity of these genes and compare the haplotype present in susceptible and resistant varieties.
3. Scientific name of the pathogens must be italic throughout the manuscript.
I hope if authors could address the above mentioned concerns it will support their study and interest the readers.
Author Response
Point to point response to reviewer 4
Dear reviewer,
We are glad to receive your valuable comments and suggestions to our manuscript. Thank you for your kind consideration on this manuscript "Screening of candidate genes associated with brown stripe resistance in sugarcane via BSR-Seq analysis". Without your professional reviews, this manuscript would not be as smooth and more persuasive as what it is now.
We have amended the manuscript according to all the opinions, suggestions and comments of the reviewers and all the changes have been marked-up in the text by the red or blue fond. The responses to all the comments and suggestions are itemized as follows:
Comment 1: Though authors investigated the gene expression profile of the resistance and susceptible line and tracked down the possible candidate genes, I did not see any analysis showing the difference in genomic content of these lines. or the genomic variation. I would suggest the authors to check all the possible genes individually and compare the polymorphism in these genes and compare between resistance and susceptible varieties.
Revision: Thanks for your valuable suggestion. In the present study, we obtained candidate regions associated with SBS resistance via BSR-Seq analysis, then functionally annotated the candidate regions, and obtained 279 candidate genes. Then, we use the CDD database and the TAIR database for in-depth annotation to check all the possible genes individually, and obtain the potential function of homologous genes (Supplementary Table S5). In the end, 39 candidate genes associated with SBS resistance were obtained. From those data presented in our manuscript, it is may not be feasible or even impossible to conduct the analysis on the difference in genomic content of these lines or the genomic variation. Regarding your suggestion that “check all the possible genes individually and compare the polymorphism in these genes and compare between resistance and susceptible varieties.”, in fact, to identify the key candidate genes related to the SBS-resistance in this study, the expression profiles of the candidate genes were analyzed based on the BSR-Seq datasets (contains YT93-159 (SBS-resistant, T01) and ROC22 (SBS-susceptible, T02), and the opposite bulks of SBS-susceptible mixed bulks (T03) and SBS-resistant mixed bulks (T04)), and the expression heat map was drawn based on the FPKM expression values. In addition, Figure 4A specifically shows the expression profile results of 39 candidate genes. Lines 235-242 have detailed description results of differential expression of candidate genes in SBS-resistant/SBS-susceptible parents and two mixed bulks. Subsequently, 20 test genes were verified by qRT-PCR. Thanks again.
Comment 2: Authors should check the haplotypic diversity of these genes and compare the haplotype present in susceptible and resistant varieties.
Revision: Thanks for your valuable comments. It is difficult to accurately analyze haplotypic diversity based on mixed ponds and parents, especially for high-ploidy crops such as sugarcane, so it is more realistic to compare the differences of SNPs among resistant/susceptible individuals. Therefore, BSR-Seq technology was finally selected. In this study, we aim to screen candidate genes associated with brown stripe resistance in sugarcane via BSR-Seq analysis. In 4.4 and 2.4, the use of GATK for the detection and filtering of single nucleotide polymorphisms (SNPs) between four samples and reference genomes is described, resulting in a high-quality, polymorphic SNP site for subsequent BSR-Seq analysis. A total of 12 SBS-resistance associated regions were identified by ED algorithm analysis (Figure 2A), covering a length of 18.46 Mb, which contained a total of 731 genes, including 24 nonsynonymous mutation genes (Supplementary Table S2). Additionally, there were other 10 SBS-resistance associated regions identified by ΔSNP-index analysis (Figure 2B). These regions covered a length of 17.84 Mb, and contained a total of 752 genes, of which 28 were nonsynonymous mutation genes (Supplementary Table S3). The results showed that the number of the associated sites obtained was more by the ED algorithm and the relative distribution density was smaller than that of the ΔSNP-index analysis. Combining the results of association analysis by ED algorithm and ΔSNP-index (Supplementary Table S4), six candidate regions associated with the SBS-resistance were finally located, which covered a length of 8.72 Mb and contain 384 annotated candidate genes. Among them, there were 17 nonsynonymous mutation SNPs between T01 vs. T02, and 48 between T03 vs. T04. Finally, the candidate gene associated with SBS resistance were obtained through gene annotation of candidate regions. We will try our best to check the haplotypic diversity of these genes and compare the haplotype present in susceptible and resistant varieties, which should no doubt be a very good and very important frontier topics in our group research. We do hope that it will be a success in the future.
Comment 3: Scientific name of the pathogens must be italic throughout the manuscript.
Revision: Thanks for your valuable comment. We revised accordingly and this is to confirm that the scientific names of the pathogens are now all italic throughout the manuscript.
Take this opportunity, again we do want to express our high and great appreciation for your kind words and professional suggestions, which should no doubt help for our future research. Thanks again.
Any questions, we will be more than happy to answer. Looking forward to hearing from you soon.
Regards and Best wishes!
Youxiong Que and Liping Xu
2022-11-24
Round 2
Reviewer 1 Report
The manuscript has been revised according to the comments, and revised manuscript has been greatly improved. The manuscript can be published.
Author Response
Dear reviewer,
Thank you for your kind consideration on this manuscript "Screening of candidate genes associated with brown stripe resistance in sugarcane via BSR-seq analysis". Without your professional reviews, this manuscript would not be as smooth and more persuasive as what it is now.
Thanks again for your kind words and professional suggestions.
Regards and Best wishes!
Youxiong Que and Liping Xu
2022-11-30
Reviewer 2 Report
Thanks for the answer. For the Comment 4: QTL mapping of F1 segregating population. The parent genome of sugarcane is heterozygous, so F1 population can be used for gene mapping. It is different with genome homozygous crops, like Arabidopsis thaliana, Cucumis sativus, and Oryza sativa etc.. I suggest that the author add some reports about crops which genomic heterozygosity, like sugarcane, can be used for gene mapping with F1 populations.
Author Response
Point to point response to reviewer 2
Dear reviewer,
We are glad to receive your valuable comments and suggestions to our manuscript. Only with your professional reviews, can this manuscript be smooth and more persuasive. The responses to your comments and suggestions are itemized as follows:
Comment 4: QTL mapping of F1 segregating population. The parent genome of sugarcane is heterozygous, so F1 population can be used for gene mapping. It is different with genome homozygous crops, like Arabidopsis thaliana, Cucumis sativus, and Oryza sativa etc.. I suggest that the author add some reports about crops which genomic heterozygosity, like sugarcane, can be used for gene mapping with F1 populations.
2nd Revision: Thanks for your valuable suggestions. According to your comments, in the section of Introduction and Discussion, we added some previous reports about QTL mapping using a F1 segregating population in sugarcane or other species, and updated references. Please see Lines 80-89 and Lines 321-328 for details. Thanks again.
Thanks again for your kind words and professional suggestions.
Regards and Best wishes!
Youxiong Que and Liping Xu
2022-11-30
Reviewer 3 Report
This revised version has addressed some of the suggestions, however, there still somme suggestions that were not adopted or improved, including Fig 2A, it is still hard to see where those 12 SBS-resistance associated regions identified by ED algorithm analysis (Fig.2A), can you clearly label othose regions?
For the original comment 2, they didn't do anything.
For comment 4, I don't think Fig. 6 is necessary, it is still there.
Author Response
Point to point response to Reviewer 3
Dear reviewer,
We are glad to receive your valuable comments and suggestions to our manuscript. Thanks again for your kind consideration on our manuscript.
We have amended the manuscript accordingly and the responses to all the comments and suggestions are itemized as follows:
Comment 1: This revised version has addressed some of the suggestions, however, there still some suggestions that were not adopted or improved, including Fig 2A, it is still hard to see where those 12 SBS-resistance associated regions identified by ED algorithm analysis (Fig.2A), can you clearly label those regions?
2nd Revision: Thanks for your valuable comment. We are very sorry that the last round of answers on this question did not give you a better understanding. Here in the second round of revision, we have provided a detailed, systematic analysis of ED algorithm in sections 2.4 and 4.4. In general, Fig. 2A just shows the fitted ED value as like the research reports of previous scholars. In order to help reader better understand the analysis of SBS resistance association regions by ED algorithm in this study, we added the Supplementary Table S6 in the last revision, and then combined with the Supplementary Table S2, we all can clearly see the distribution positions of 12 association regions on chromosomes and the fitted ED values of association sites (including the start position and end position, the number of associated regions on a chromosome, and ED values for all SNP sites). Based on the references we have read, no researcher has label these regions on this figure especially in crops with extremely complex genetic backgrounds, because of there are thousands of SNP sites on a chromosome corresponding to ED values. Therefore, we can't label 12 SBS-resistance association regions on Fig. 2A. We hope to get your understanding. In addition, in order to show Figs. 2A and 2B more clearly, we split Fig. 2A/2B into Fig. 2 and Fig. 3, and we have made minor modifications to these figures (Lines 191-202). We especially hope that this time, our responses is clear and understandable. Below are some related research articles related to this comment:
https://www.frontiersin.org/articles/10.3389/fpls.2022.1035266/full. https://acsess.onlinelibrary.wiley.com/doi/10.1002/tpg2.20120. https://link.springer.com/article/10.1007/s00122-019-03419-9. https://www.frontiersin.org/articles/10.3389/fpls.2021.764978/full.
Comment 2: After the identification of the DEGs and the genomic regions for SBS resistance. It should be focused to validate the regions harboring SBS resistance, not to validate the expression level of those so-called candidate genes. I strongly suggest to add the validation of those regions.
2nd Revision: Thanks for your valuable comment. From RNA-seq to association analysis until SBS resistance association regions (six candidate regions associated with the SBS resistance were finally located, which covered a length of 8.72 Mb and contained 279 annotated candidate genes) were obtained. According to your comment, we added new gene analysis (Lines 523-526; Lines 144-146), DEGs analysis of SBS resistance association regions (Lines 579; Lines 237-243; supplementary Table S6), COG classification statistical analysis (Lines 243-249; supplementary Figure S2) in the current manuscript. Then, 39 candidate genes closely linked to the target trait were identified in SBS resistance association region by in-depth annotated. It is very important to find closely linked resistance genes in the the regions harboring SBS resistance. Through RNA-seq datasets, we analyzed the expression profiling of candidate genes in the associated region in resistant/susceptible parents, and randomly selected 20 genes in the SBS resistance associated region to verify their expression patterns by qRT-PCR. We are now more confident that the present version of this manuscript should be a relatively complete story of this study. Thanks again for your useful comments and suggestions, and during the two rounds of review, they have been incorporated into the revisions on the introduction, results, discussion, materials and methods, and conclusions. We do hope that the above response is relatively clear and can satisfy your comment.
Comment 4: I don't think Fig. 6 is necessary, it is still there.
2nd Revision: Thanks for this suggestion. According to your suggestion, we have moved Fig. 6 to the supplementary files (Supplementary Fig. S4). Thanks again.
Any questions, we will be more than happy to answer. Looking forward to hearing from you soon.
Regards and Best wishes!
Youxiong Que and Liping Xu
2022-11-30
Reviewer 4 Report
Dear Authors,
Thanks for addressing the comments.
It can be accepted in current form.
Author Response

(The authors gave the same response as above.)

Round 3
Reviewer 3 Report
All my concerns have been addressed in this revised manuscript and the quality of the ms has been improved.